# Factors associated with poor adherence to intrapartum fetal heart monitoring in relationship to intrapartum related death: A prospective cohort study

**Annette Ekblom**[1], **Mats Målqvist**[1], **Rejina Gurung**[1], **Angela Rossley**[1], **Omkar Basnet**[2], **Pratiksha Bhattarai**[2], **Ashish K. C.**[1,3]*

**1** Uppsala Global Health Research on Implementation and Sustainability (UGHRIS), Department of Women's and Children's Health, Uppsala University, Uppsala, Sweden, **2** Golden Community, Lalitpur, Nepal, **3** Society of Public Health Physicians Nepal, Kathmandu, Nepal

* ashish.k.c@kbh.uu.se

## Abstract

### Background

Poor quality of intrapartum care remains a global health challenge for reducing stillbirth and early neonatal mortality. Despite fetal heart rate monitoring (FHRM) being key to identify fetus at risk during labor, sub-optimal care prevails in low-income settings. The study aims to assess the predictors of suboptimal fetal heart rate monitoring and assess the association of sub-optimal FHRM and intrapartum related deaths.

### Method

A prospective cohort study was conducted in 12 hospitals between April 2017 to October 2018. Pregnant women with fetal heart sound present during admission were included. Inferential statistics were used to assess proportion of sub-optimal FHRM. Multi-level logistic regression was used to detect association between sub-optimal FHRM and intrapartum related death.

### Result

The study cohort included 83,709 deliveries, in which in more than half of women received suboptimal FHRM (56%). The sub-optimal FHRM was higher among women with obstetric complication than those with no complication (68.8% vs 55.5%, p-value<0.001). The sub-optimal FHRM was higher if partograph was not used than for whom partograph was completely filled (70.8% vs 15.9%, p-value<0.001). The sub-optimal FHRM was higher if the women had no companion during labor than those who had companion during labor (57.5% vs 49.6%, p-value<0.001). After adjusting for background characteristics and intra-partum factors, the odds of intrapartum related death was higher if FHRM was done sub-optimally in reference to women who had FHRM monitored as per protocol (aOR, 1.47; 95% CI; 1.13, 1.92).

**Data Availability Statement:** All relevant data are within the paper and its Supporting Information files.

**Funding:** The authors received no specific funding for this work.

**Competing interests:** The authors have declared that no competing interests exist.

## Conclusion

Adherence to FHRM as per clinical standards was inadequate in these hospitals of Nepal. Furthermore, there was an increased odds of intra-partum death if FHRM had not been carried out as per clinical standards. FHRM provided as per protocol is key to identify fetuses at risk, and efforts are needed to improve the adherence of quality of care to prevent death.

## Introduction

Every year around the globe, an estimated 1.0 million stillbirths take place during labour (intrapartum period) [1, 2] and 0.7 million neonates die during the first day of birth [3, 4]. Almost 10 million neonates do not cry at birth [5] and 2 million neonates have delayed developmental milestones and disability due to intrapartum related complication [6]. Ninety-eight percent of these deaths and disability cases take place in low- and middle-income settings, especially in South Asia and Sub-saharan Africa, where women cannot access high quality intrapartum care [4, 7, 8]. Failure to detect high-risk fetuses during labour and intervene accordingly is one of the major attributors to intrapartum related death leading to neonates requiring additional interventions at birth and later in life [9, 10].

To address the burden and improve the care for mothers and newborns, the World Health Assembly in 2014 endorsed the Every Newborn Action Plan [10]. The Every Newborn Action plan lays out several key strategic interventions to reduce mortality and morbidity during labour, childbirth and the postnatal period. One of the key strategic interventions is to improve the quality of intrapartum care during labour and childbirth [11–13].

Fetal heart rate monitoring (FHRM) during labour is an effective intrapartum intervention [14] to detect acute or prolonged stress to the fetus, clinically manifested as deceleration or acceleration of the fetal heart rate [15–17]. A prolonged abnormal fetal heart rate will cause perinatal depression, stillbirth, neonatal death and neonatal encephalopathy [18]. An early intervention such as intrapartum resuscitation, instrumental delivery or emergency cesarean section can prevent adverse birth outcomes.

Despite the availability of the World Health (WHO) standards for fetal heart rate monitoring during labour and childbirth [19, 20], the health system barriers exist preventing adherence to clinical standards in low resource settings [21]. These health system barriers include lack of availability of fetal heart rate monitors, poor competency to use FHRM, lack of clinical decision support system and adequate human resource [22–24].

A multi-country study on the health system barriers for maternal and newborn care in South Asia and Africa has shown that governance, financing and health workers competency are the major barriers for labour monitoring [25]. Despite the availability of different quality improvement approaches to overcome the health system barriers [26, 27], the coverage of fetal heart rate monitoring as per the WHO's standard remains low.

Studies from public hospitals in Nepal have shown that the adherence to FHRM and partograph use was inadequate [28–30]. Further, inadequate infrastructure, poor supply of essential newborn commodities, ineffective clinical supervision and mentorship for service providers are significant barriers and essentially needed to be addressed in order to ensure effectiveness in newborn interventions [28]. This study aims to assess the predictors of suboptimal fetal heart rate monitoring and assess the association of sub-optimal FHRM and intrapartum related deaths.

## Materials and methods

### Study design

This was a prospective cohort study (S1 Checklist), nested within a large stepped-wedge cluster-randomized controlled trial, Nepal Perinatal Quality Improvement Project (NePeriQIP) registration number ISRCTN30829654 [31]. In each wedge a high volume (>6000 annual birth), a medium volume (1500–6000 annual birth) and a low volume (<1500 annual birth) hospital were allocated. The 12 public hospitals were selected to represent the public referral hospital in each province of the country. The data was collected over an 18-month time frame between April 2017 and October 2018.

### Study setting

The study hospitals had annual deliveries of approximately 1,500 to 11,000, and an estimated intrapartum mortality rate of 20 per 1000 births. All the hospitals provided normal vaginal, assisted vaginal and caserean section services. The low volume hospitals (Bardiya, Pyuthan, Nuwakot, and Prithivi Chandra (Nawalparasi) did not have specialized newborn care units, while the high-volume hospitals (Koshi Zonal, Bharatpur, Lumbini Zonal, and Bheri Zonal) as well as the medium volume hospitals (Western regional, Rapti Sub-regional, Mid- Western regional, and Seti Zonal) provided comprehensive specialized newborn care. Skilled birth attendants and midwives led the labour units, while the pediatricians led the neonatal care units for sick newborns. All deliveries had access to neonatal resuscitation at birth.

### Participants

Woman with fetal heart sound (FHS) present during the admission, gestational age of $\geq 22$ weeks or a fetal birth weight of $\geq 500$ grams were eligible for the study. Women with absence of FHS upon admission and antepartum stillbirth were excluded. All women who consented to participate in the study were enrolled and were followed up upon discharge from the hospitals.

### Data collection and management

A surveillance system consisting of independent research nurses were established in all the hospitals [32]. These research nurses were supervised by a research nurse coordinator in each hospital. The research nurses were stationed in the labour and delivery and postnatal units all around the clock throughout the study period. After the woman was admitted in the labour room, she was assessed for eligibility by the research nurse. Woman who was eligible for the study were approached by the research nurses to be enrolled in the study. Woman enrolled in the study were followed from the time of admission until discharge from the hospital. The research nurses collected the women's medical information from medical records and socio-economic information through semi-structured interviews at discharge in a paper-based form. The research team extracted the clinical information on fetal heart rate monitoring, obstetric progress, and birth outcome of each enrolled participants at the labour and postnatal unit. The social and demographic information was collected from semi-structured interview. The research nurse coordinator assessed the quality of data collection and ensured the completeness of data collected.

During the weekdays, the research nurse coordinator from each study site, compiled all the filled-up forms and sealed them in a closed envelop and sent it through a designated courier service to the central research office in Kathmandu. The received forms from each site were assessed for completeness by a data management team at the central research office in

Kathmandu. Followed the assessment, the paper-based forms were entered into the database by the data entry and management team. All the entered paper forms were indexed according to hospital. The database manager cleaned all the entered forms using the Survey Processing System (CS-Pro) on a monthly basis.

## Measurement

**Outcome measure.** *Intrapartum related death* was defined as the intrapartum stillbirth (>22-week stillbirth during labour) and first day neonatal death [33].

**Exposure measure.** *Adherence to FHRM protocol* during labour was deemed to be adequate when the FHR was monitored as per protocol at least every 30 minutes during the first stage of labour and at least every 15 minutes during the second stage of labour according to WHO standards [34].

**Obstetric measure.** *Number of antenatal care visits (ANC)* was the number of times a woman had received care from any provider during pregnancy. The visits were categorized to $\geq$ 4 visits, or $\leq$ 3 visits [34].

*Parity* was the number of previous pregnancies and was categorized as follows: multipara indicated two or more previous births, primipara indicated one previous birth, and nullipara were women who had never carried a pregnancy.

*Fetal heart sound (FHS) on admission* was categorized as normal (110–160 BPM) or abnormal (< 110 BPM or > 160 BPM). Mode of delivery was categorized as spontaneous vaginal, assisted vaginal, and caesarean section (CS) [34].

*Companion present during labour and childbirth* was categorized as pregnant women had a family members present during labour and childbirth [35].

*Birth weight* the first measurement of the neonate's body weight and was categorized $\geq$ 2500 grams, and < 2500 grams (low birth weight) [36].

*Gestational age* was calculated using the first day of the last normal menstrual period (if known), and was categorized according to $\geq$ 37 weeks, and < 37 weeks (pre-term birth) [37].

*Obstetric complication after admission [34]* indicated one or more of the following complication: antepartum hemorrhage, decreased fetal movements, prolonged labour (more than 12 hours in the active phase), pre-eclampsia (blood pressure greater than 140/90 mmHg and positive proteinuria), eclampsia (including convulsions, coma, stroke or unconsciousness), breech or transverse lie, prolapsed cord, chorio-amnionitis, premature rupture of membrane, preterm labour, and/or fetal congenital anomaly.

*Fetal distress* in labour takes place when the fetus was not well, commonly due to fetal hypoxia with decreased fetal movement [34].

*Prolonged labour* was when the cervix was not dilated more than 4cm after eight hours of regular contractions, or if cervical dilatation was to the right of the alert line on the partograph [34].

*Pre-term* birth was defined when the neonate was born before 37 completed weeks of gestation, as estimated by the data of the mother's last menstrual period or based on clinical examination of the newborn [34].

## Sample size

In the NePeriQIP data set, 87,989 women were considered according to their birth records at the 12 public hospitals in Nepal (S1 Dataset). After screening on exclusion criterion and handling of missing values, a total of 83,709 women remained in the study (Fig 1).

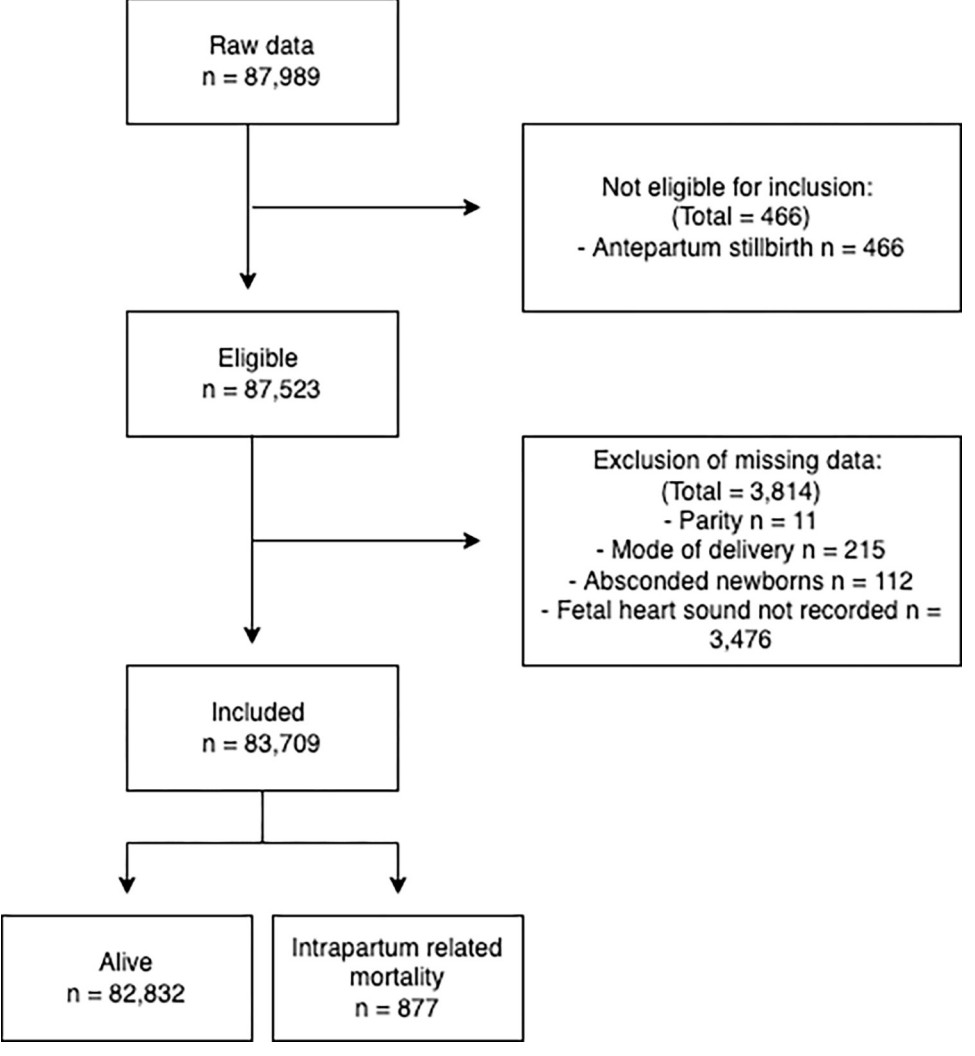

**Fig 1. Flow chart showing the study participants.**

## Statistical analysis

We made the comparison on the coverage of standard and sub-standard FHRM by parity, number of ANC visit, obstetric complication after admission, fetal heart rate at admission, companion during labour and childbirth, fetal distress in labour, use of partograph, multiple delivery and mode of birth using Pearson's chi-square test.

The bi-variate association between intrapartum related death with primary exposure- FHRM and other exposures (uneducated women, ethnicity, maternal age, parity, number of ANC visit, complication during admission, multiple delivery, mode of birth, low birth weight, pre-term birth and sex of baby) was assessed using logistic regression.

A multi-level association was assessed between intrapartum related death with the primary exposure and other exposures. Three different modeling was conducted to assess the association. For model I, demographic and antenatal variables were adjusted. Demographic and antepartum variables were maternal education, maternal ethnicity, maternal age, parity, antenatal check-up from skilled provider. For model II, intrapartum variables were adjusted. Intrapartum variables were obstetrics complication during labour, multiple pregnancy, mode of birth,

preterm birth, low birth weight and sex of baby. For model III, demographic, antenatal and intrapartum variables were adjusted. The statistical software program used was R version 1.4.11.03, and the significance level was set to p = 0.05.

The approach of simple imputation was used for the values missing at random (MAR) of birth weight and gestational age and used each other as predictor variables which was done based on a strong correlation between the two variables (p<0.001). Five different data sets were generated with the predictive mean matching (pmm) method through the MICE (multi-variate imputation by chained equations) method. A sensitivity analysis using the proportion of low birth weight (LBW) as a proxy was conducted to determine the dataset that best fitted the original data.

The study was approved by Ethical Review Board of Nepal Health Research Council (reference number 26–2017). Written informed consent was obtained from the women before inclusion in the NePeriQIP study and confidentiality was maintained.

## Results

During the study period a total of 87,989 women came to the 12 hospitals for delivery and were included in the NePeriQIP data collection, of which 83,709 women were selected for inclusion in the current study. A total of 209 intrapartum stillbirths and 668 early neonatal deaths during the study period, with intrapartum related mortality rate of 10.5 per thousand deliveries (Fig 1). The probability of intrapartum related mortality varied by hospital on a scale from 5–26 per 1000 birth (Fig 2). Of the 12 hospitals, more than 75% of the women delivering in Lumbini and Bheri hospital had sub-optimal FHRM. Almost 20% of women delivering in Bardiya and Mid-western hospitals had sub-optimal care (Fig 3).

The proportion of sub-optimal FHRM was higher among the multiparous women than nulliparous women (58.0% vs 54.2%, p-value<0.001). The proportion of sub-optimal FHRM was higher among women with obstetric complication than those with no complication (68.8% vs 55.5%, p-value<0.001). The proportion of sub-optimal FHRM was higher if the women had no companion during labour than those who had companion during labour (57.5% vs 49.6%, p-value<0.001). The proportion of sub-optimal FHRM was higher if the women had preterm labour than those who did not have preterm labour (75.3% vs 55.7%, p-value<0.001). The proportion of sub-optimal FHRM was higher if partograph was not used than for whom partograph was completely filled (70.8% vs 15.9%, p-value<0.001). The proportion of sub-optimal FHRM was higher with multiple pregnancy than with women who had singleton pregnancy (61.1% vs 55.7%, p-value = 0.002). The proportion of sub-optimal FHRM was higher if women had cesarean birth than women who had spontaneous birth (67.0% vs 52.6%, p-value<0.001) (Table 1).

In the bi-variate analysis to assess the association between sub-optimal FHRM and intrapartum related deaths, the crude risk of intrapartum related death was 1.34 times higher if there was no FHRM as per protocol in reference to those who had FHRM as per protocol (cOR, 1.34; 95% CI; 1.17, 1.53). The crude risk of intrapartum related death was 2.60 times higher if the women were uneducated in reference to those who were educated (cOR, 2.60; 95% CI; 2.00, 3.40). The crude risk of intrapartum related death was 1.84 times higher if women were from Muslim ethnic group in reference to those who were from Chettri-brahmin (cOR, 1.84; 95% CI; 1.31, 2.60). The crude risk of intrapartum related death was 2.24 times higher if the women were 36 years and above in reference to those who had age 26–30 years of age (cOR, 2.24; 95% CI; 1.46, 3.43). The crude risk of intrapartum related death was 2.04 times higher if the women had 3 or less ANC visit than those who had 4 or more ANC visit (cOR, 2.04; 95% CI; 1.58, 2.15). The crude risk of intrapartum related death was 4.53 times higher if the women

**Fig 2. Point estimate of intrapartum related death by hospital.**

had obstetric complication in reference to those who had no obstetric complication (cOR, 4.53; 95% CI; 1.58, 2.63). The crude risk of intrapartum related death was 6.08 times higher if the women had multiple pregnancy in reference to women with singleton pregnancy (cOR, 6.08; 95% CI; 4.54, 8.13). The crude risk of intrapartum related death was 5.42 times higher if the neonates had low birth weight in reference to the neonates with normal birth weight (cOR, 5.42; 95% CI; 4.69, 6.26). The crude risk of intrapartum related death was 4.43 times higher if the neonates were born pre-term in reference to the neonates born term (cOR, 5.42; 95% CI; 4.69, 6.26) (Table 2).

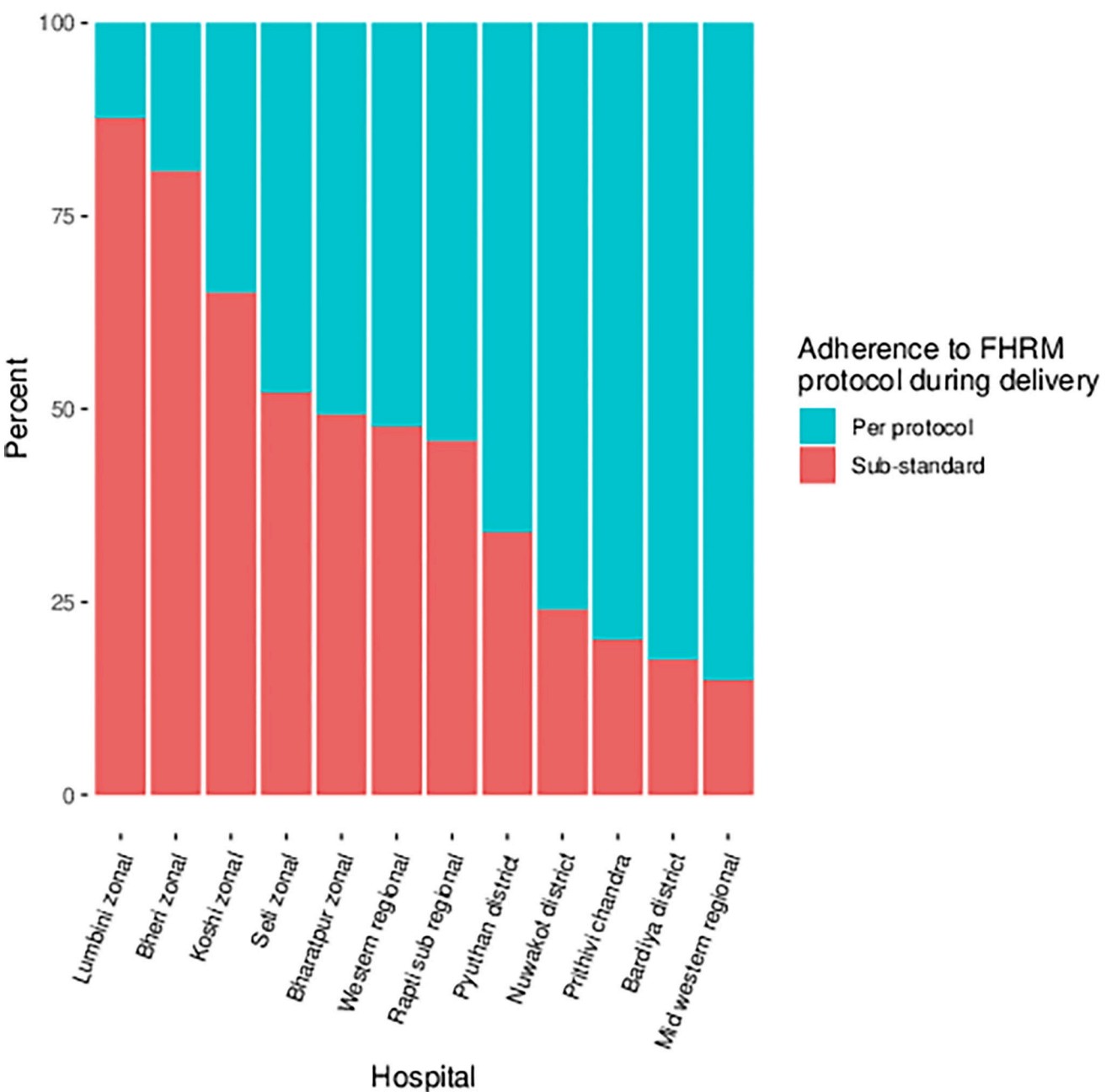

**Fig 3. Coverage of FHRM protocol as per protocol during labor by hospital.**

In the multi-level analysis to assess the risk of intrapartum related death with sub-optimal FHRM, three different adjustment modeling were done. In model I, after adjusting with antenatal factors (maternal education, ethnicity, maternal age, parity and 4 or more ANC check-up), the risk of intrapartum related death was 1.52 higher if FHRM was done sub-optimally in reference to women who had FHRM monitored as per protocol (aOR, 1.52; 95% CI; 1.17, 1.97). In model II, after adjusting with intrapartum factors (obstetric complication, multiple pregnancy, mode of birth, preterm birth, low birth weight and sex of neonate), the risk of intrapartum related death was 1.30 higher if FHRM was done sub-optimally in reference to

**Table 1.  Coverage to FHRM protocol (n = 83709).**

|  | FHRM as per protocol | | |
|---|---|---|---|
|  | Yes, 37011 (44.2%) | No, 46698 (55.8%) | p-value |
| Parity |  |  |  |
| Nullipara (38,698) | 17719 (45.8%) | 20970 (54.2%) | <0.001 |
| Primipara (29,380) | 12352 (44.4%) | 17028 (55.6%) |  |
| Multipara (15,640) | 6940 (42.0%) | 8700 (58.0%) |  |
| Number of ANC visit |  |  |  |
| <4 visit (14,302) | 6199 (44.2%) | 8103 (55.8%) |  |
| ≥4 visit (50,067) | 22117 (43.3%) | 27950 (56.7%) | 0.079 |
| Complication during Admission |  |  |  |
| No (81,658) | 36371 (44.5%) | 45287 (55.5%) |  |
| Yes (2051) | 640 (31.2%) | 1411 (68.8%) | <0.001 |
| Fetal heart rate at admission |  |  |  |
| Normal (83,519) | 36927 (44.2%) | 46592 (55.8%) |  |
| Abnormal (190) | 84 (44.2%) | 106 (55.8%) | 0.999 |
| Companion during labor |  |  |  |
| No (53,216) | 22623 (42.5%) | 30593 (57.5%) |  |
| Yes (11,740) | 5922 (50.4%) | 5818 (49.6%) | <0.001 |
| Fetal distress in labor |  |  |  |
| No (82,724) | 36607 (44.3%) | 46117 (55.7%) |  |
| Yes (985) | 404 (41.0%) | 581 (59.0%) | 0.042 |
| Pre-term labor |  |  |  |
| No (83,422) | 36940 (44.3%) | 46482 (55.7%) |  |
| Yes (287) | 71 (24.7%) | 216 (75.3%) | <0.001 |
| Prolonged labor |  |  |  |
| No (83,560) | 36956 (44.2%) | 46604 (55.8%) |  |
| Yes (149) | 55 (36.9%) | 94 (63.1%) | 0.083 |
| Partograph use |  |  |  |
| Yes, completely filled (23,201) | 19513 (84.1%) | 3688 (15.9%) | <0.001 |
| Yes, partially filled (23,542) | 6710 (28.5%) | 16832 (71.5%) |  |
| Not recorded (36,966) | 10788 (29.2%) | 26178 (70.8%) |  |
| Multiple delivery |  |  |  |
| No (82,825) | 36667 (44.3%) | 46158 (55.7%) |  |
| Yes (884) | 344 (38.9%) | 540 (61.1%) | 0.002 |
| Mode of birth |  |  |  |
| Spontaneous vaginal (62,108) | 29436 (47.4%) | 32672 (52.6%) | <0.001 |
| Assisted vaginal (3,341) | 1542 (46.2%) | 1799 (53.8%) |  |
| Cesarean birth (18,260) | 6033 (33.0%) | 12227 (67.0%) |  |

women who had FHRM monitored as per protocol (aOR, 1.30; 95% CI; 1.11, 1.51). In model III, after adjusting with antenatal and intrapartum factors, the risk of intrapartum related death was 1.47 higher if FHRM was done sub-optimally in reference to women who had FHRM monitored as per protocol (aOR, 1.47; 95% CI; 1.13, 1.92) (Table 3).

## Discussion

This study showed that only half of the women have their fetal heart rate monitored as per protocol. The proportion of sub-optimal FHRM was high if women did not have companion present during labour. Women who had obstetric complication after admission had lower FHRM

**Table 2. The bi-variate association between FHRM and intrapartum related death (n = 83709).**

| | Alive (82,832, 99.0%) | Intrapartum related death (877, 1.0%) | cOR, 95% CI |
|---|---|---|---|
| FHHRM as per protocol | | | |
| Yes (37,011) | 36684 (99.1%) | 327 (0.9%) | |
| No (46,698) | 46148 (98.8%) | 550 (1.2%) | 1.34 (1.17, 1.53) |
| Uneducated women | | | |
| No (55,817) | 55633 (99.7%) | 184 (0.3%) | |
| Yes (9,139) | 9061 (99.1%) | 78 (0.9%) | 2.60 (2.00, 3.40) |
| Ethnicity | | | |
| Dalit (13,018) | 12849 (98.7%) | 169 (1.3%) | 1.49 (1.23, 1.81) |
| Janajati (24,632) | 24376 (99.0%) | 256 (1.0%) | 1.19 (1.01, 1.41) |
| Madhesi (6,925) | 6852 (98.9%) | 73 (1.1%) | 1.21 (0.93, 1.56) |
| Muslim (2,314) | 2277 (98.4%) | 37 (1.6%) | 1.84 (1.31, 2.60) |
| Chhetri/Brahmin (32,932) | 32644 (99.1%) | 288 (0.9%) | |
| Others (3,888) | 3834 (98.6%) | 54 (1.4%) | 1.60 (1.19, 2.14) |
| Maternal age | | | |
| 18 or less yrs (6353) | 6275 (98.8%) | 78 (1.2%) | 1.20 (0.92, 1.56) |
| 19 to 25 yrs (51,627) | 51111 (99.0%) | 516 (1.0%) | 0.98 (0.83, 1.15) |
| 26–30 yrs (19,604) | 19403 (99.0%) | 201 (1.0%) | |
| 31 to 35 yrs (5,065) | 5007 (98.9%) | 58 (1.1%) | 1.12 (0.83, 1.50) |
| 36 and more yrs (1,060) | 1036 (97.7%) | 24 (2.3%) | 2.24 (1.46, 3.43) |
| Parity | | | |
| Nullipara (38,689) | 38,358 (99.1%) | 331 (0.9%) | 0.90 (0.77, 1.06) |
| Primipara (29,380) | 29101 (99.1%) | 279 (0.9%) | |
| Multipara (15,640) | 15373 (98.3%) | 267 (1.7%) | 1.81 (1.53, 2.15) |
| Number of ANC visit | | | |
| 3 or less ANC visit (14,302) | 14208 (99.3%) | 94 (0.7%) | |
| 4 or more ANC visit (50,067) | 49905 (99.7%) | 162 (0.3%) | 2.04 (1.58, 2.63) |
| Obstetric complication during Admission | | | |
| No (81,658) | 80868 (99.2%) | 790 (0.97%) | |
| Yes (2,051) | 1964 (95.8%) | 87 (4.2%) | 4.53 (3.62, 5.68) |
| Multiple delivery | | | |
| No (82,825) | 81999 (99.0%) | 826 (1.0%) | |
| Yes (884) | 833 (94.2%) | 51 (5.8%) | 6.08 (4.54, 8.13) |
| Mode of birth | | | |
| Spontaneous vaginal (62,108) | 61494 (99.0%) | 614 (1.0%) | |
| Assisted vaginal (3,341) | 3235 (96.8%) | 106 (3.2%) | 3.28 (2.66, 4.04) |
| Caesarean birth (18,260) | 18103 (99.1%) | 157 (0.9%) | 0.87 (0.73, 1.04) |
| Low Birth Weight | | | |
| Low Birth Weight (13,669) | 13268 (97.1%) | 401 (2.9%) | 5.42 (4.69, 6.26) |
| Normal birth weight (64,199) | 63843 (99.4%) | 356 (0.6%) | |
| Preterm birth | | | |
| Preterm birth (14,431) | 14055 (97.4%) | 376 (2.6%) | 4.43 (3.84, 5.11) |
| Term birth (63,437) | 63056 (99.4%) | 381 (0.6%) | |
| Sex of baby | | | |
| Male (45,276) | 44770 (98.9%) | 506 (1.1%) | |
| Female (38,433) | 38062 (99.0%) | 371 (1.0%) | 1.16 (1.01, 1.33) |

cOR = Crude Odds Ratio.

CI = Confidence Interval.

**Table 3. Multi-level analysis and modeling to assess the association between FHRM as per protocol and intrapartum related death.**

| | p-value | Model I; aOR, 95% CI | p-value | Model II; aOR, 95% CI | p-value | Model III; aOR, 95% CI |
|---|---|---|---|---|---|---|
| FHRM as per protocol | | | | | | |
| Yes | | Reference | | Reference | | Reference |
| No | 0.002 | 1.52 (1.17, 1.97) | 0.001 | 1.30 (1.11, 1.51) | 0.005 | 1.47 (1.13, 1.92) |
| Maternal education | | | | | | |
| Yes | | Reference | | | | Reference |
| No | <0.001 | 2.04 (1.52, 2.74) | | | <0.001 | 1.85 (1.37, 2.51) |
| Ethnicity | | | | | | |
| Chhetri/Brahmin | | Reference | | | | Reference |
| Dalit | 0.253 | 1.24 (0.86, 1.78) | | | 0.327 | 1.20 (0.83, 1.75) |
| Janajati | 0.28 | 1.20 (0.86, 1.66) | | | 0.197 | 1.24 (0.89, 1.74) |
| Madhesi | 0.553 | 1.14 (0.73, 1.78) | | | 0.587 | 1.14 (0.70, 1.80) |
| Muslim | 0.253 | 1.41 (0.78, 2.56) | | | 0.398 | 1.32 (0.63, 2.15) |
| Others | 0.709 | 1.12 (0.61, 2.07) | | | 0.636 | 1.16 (0.63, 2.15) |
| Maternal age | | | | | | |
| 26–30 yrs | | Reference | | | 0.013 | Reference |
| 18 or less yrs | 0.045 | 1.81 (1.01, 3.24) | | | 0.012 | 2.13 (1.18, 3.85) |
| 19 to 25 yrs | 0.006 | 1.59 (1.14, 2.22) | | | 0.006 | 1.61 (1.15, 2.26) |
| 31 to 35 yrs | 0.898 | 1.04 (0.59, 1.83) | | | 0.75 | 0.91 (0.51, 1.64) |
| 36 and more yrs | 0.396 | 0.54 (0.13, 2.24) | | | 0.325 | 0.49 (0.12, 2.04) |
| Parity | | | | | | |
| Primipara | | Reference | | | | Reference |
| Nullipara | 0.01 | 0.66 (0.48, 0.91) | | | <0.001 | 0.55 (0.40, 0.76) |
| Multipara | 0.006 | 1.58 (1.14, 2.18) | | | 0.005 | 1.60 (1.15, 2.23) |
| 4 or more ANC check up | | | | | | |
| Yes | | Reference | | | | Reference |
| No | <0.001 | 1.70 (1.31, 2.20) | | | 0.001 | 1.55 (1.18, 2.02) |
| Obstetric complication during admission | | | | | | |
| No | | | | Reference | | Reference |
| Yes | | | <0.001 | 3.80 (2.95, 4.90) | 0.099 | 1.69 (0.91, 3.16) |
| Multiple pregnancy | | | | | | |
| No | | | | Reference | | Reference |
| Yes | | | <0.001 | 3.19 (2.32, 4.37) | <0.001 | 3.46 (2.00, 5.99) |
| Mode of birth | | | | | | |
| Spontaneous vaginal birth | | | | Reference | | Reference |
| Instrumental birth | | | 0.114 | 1.17 (0.96, 1.41) | <0.001 | 4.07 (2.62, 6.33) |
| Casarean birth | | | <0.001 | 4.20 (3.20, 5.51) | 0.186 | 0.80 (0.58, 1.11) |
| Preterm birth | | | | | | |
| No | | | | Reference | | Reference |
| Yes | | | <0.001 | 1.80 (1.46, 2.21) | 0.002 | 1.76 (1.22, 2.53) |
| Low birth weight | | | | | | |
| No | | | | Reference | | Reference |
| Yes | | | <0.001 | 3.32 (2.69, 4.09) | 0.002 | 1.79 (1.24, 2.60) |
| Sex of baby | | | | | | |
| Girls | | | | Reference | | Reference |
| Boy | | | <0.001 | 1.36 (1.18, 1.58) | <0.001 | 1.63 (1.25, 2.12) |

cOR = Crude Odds Ratio.

aOR = Adjusted Odds Ratio.

CI = Confidence Interval.

Model I- demographic and antepartum related variable; model II- intrapartum related variables and model III- antepartum and intrapartum related variables.

as per protocol. The risk of intrapartum related mortality increased by 47% if FHRM was not monitored as per protocol after adjusting for all demographic, antenatal and intrapartum confounders.

Inadequate adherence to standardized protocols for monitoring the intrapartum period has been a challenge in the context of Nepal, as reported in previous findings on the adherence to FHRM and partograph use [28, 30]. This study showed a positive association between presence of companion during labour and fetal heart rate monitoring. Studies have shown that women with companion during labour have better support and care [38, 39]. A companion during labour also prompted health worker for better care and attention to the women [40]. Especially in resource limited settings where health workers are scarce and over-burdened, a companion present during labour can support for intermittent FHRM [41].

Further, we also found that women who had obstetric complication after admission had higher proportion of sub-optimal of FHRM. Women with obstetric complication are high-risk pregnancies who need continuous fetal heart rate surveillance. The reasons may be a lack of staff following a heavy workload, scarcity of measuring devices, poor knowledge or inadequate understanding of the importance FHRM especially for high-risk pregnancy [42, 43]. The lack of adequate skilled staff and resources are well-known barriers which are prevalent in many LMICs and may hinder timely and accurate FHRM.

A substantial result from thus study was that suboptimal use of FHRM for monitoring the intrapartum period during labour, is associated with intrapartum related mortality. Further, the adherence to standard FHRM protocol at the different hospitals varied which impacted the intrapartum related outcome. Inadequate fetal heart monitoring coupled with obstetric complication are strong predictors for intrapartum related mortality. A potential reason is that women with obstetric complication have fetal distress, and lack of monitoring of these highly distressed fetuses led to severe fetal compromise during the intrapartum period. This further highlights the need to provide more care to vulnerable groups. In line with Hart's law of inverse care, the lack of access to high quality care to women with obstetric complication leads to intrapartum related mortality [44, 45]. Women with obstetric complication are at higher risk of poor birth outcomes and standard fetal heart rate is inversely available with these population [44, 45].

Improving the adherence to fetal heart rate monitoring during labour in these hospitals in Nepal, despite a national standard for care will require a quality improvement implementation approach. A contextual analysis in each hospital on bottlenecks for adherence to care, followed by continuous planning, implementation and review of progress is needed. Furthermore, the hospital system needs improvement in timely decision for instrumental or cesarean section for high-risk fetuses.

A strength of the current study is the large sample size of 83,000 women and the diverse settings of 12 public hospitals distributed all over the country, making generalizability plausible, however, not stark. The hospitals were representative of different levels of the public health system of Nepal. In this study, by combining intrapartum stillbirths with neonatal deaths to intrapartum related mortality, the risk of misclassification was reduced.

There were some limitations in this study. First, this is based on the extraction of the data from medical records, so there might be some error in recording the clinical events. Second, the probability of intrapartum death varied by hospital indicating the difference in the quality of intrapartum care, so we did not assess the heterogeneity of the care. Third, handling of missing data with a simple imputation method may have introduced bias with an exaggeration or attenuation of the association of interest, however, a correction with a sensitivity analysis was performed. Finally, the effect of potential confounders such as length of time from admission to delivery was not adjusted for because the data set lacked this information.

## Conclusion

FHRM detects high risk women with an abnormal fetal heart rate and can thus trigger early intervention for childbirth. This study adds that women who received sub-standard FHRM had higher risk of intrapartum related mortality. This study highlights that FHRM was inadequate in public hospitals in Nepal, and suboptimal care is higher among those who are at risk. As the demand for facility-based deliveries increases in Nepal, providing quality of intrapartum care that meet the population demand is essential. High quality intrapartum care, such as fetal heart rate monitoring with timely action for high-risk fetus and mother, will prevent intrapartum mortality.

## Supporting information

**S1 Checklist. STROBE statement checklist- Factors associated with poor adherence to intrapartum fetal heart monitoring in relationship to intrapartum related death: A prospective cohort study.**
(DOCX)

**S1 Dataset.**
(CSV)

## Acknowledgments

Ongoing research assistance was provided by Kasperi Kilpi and Angela Rossley. Statistical revisions were recommended by Anna Wikman. We would like to thank all the data coordinators and surveillance officers, hospital manager, nursing in-charge, all the nurses and support staff working in the sick newborn care units. We thank all the mothers and caregivers of the sick newborns who consented to the study.

## Author Contributions

**Conceptualization:** Pratiksha Bhattarai, Ashish K. C.

**Data curation:** Ashish K. C.

**Investigation:** Annette Ekblom, Mats Målqvist, Rejina Gurung, Angela Rossley, Pratiksha Bhattarai, Ashish K. C.

**Methodology:** Annette Ekblom, Angela Rossley.

**Project administration:** Rejina Gurung, Omkar Basnet, Pratiksha Bhattarai.

**Software:** Angela Rossley.

**Supervision:** Mats Målqvist, Rejina Gurung, Omkar Basnet, Pratiksha Bhattarai, Ashish K. C.

**Writing – original draft:** Annette Ekblom.

**Writing – review & editing:** Mats Målqvist, Rejina Gurung, Angela Rossley, Omkar Basnet, Pratiksha Bhattarai, Ashish K. C.

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
