## [Decision Letter · Decision Letter 0]

18 Nov 2021

PGPH-D-21-00535

Association of poor adherence to intrapartum fetal heart rate monitoring with intrapartum related death: a prospective cohort study

Dear Dr. KC,

Thank you for submitting your manuscript to PLOS Global Public Health. After careful consideration, we feel that it has merit but does not fully meet PLOS Global Public Health’s publication criteria as it currently stands. Therefore, we invite you to submit a revised version of the manuscript that addresses the points raised during the review process.

This is an important topic and I would like to see this published, however a number of the Reviewers have raised concerns or questions that need to be addressed. Please see additional Editor comments, and Reviewer comments below.

We look forward to receiving your revised manuscript.

Kind regards,

Hamish R. Graham, PhD

Academic Editor

Journal Requirements:

1. Please ensure you have included the registration number for the clinical trial referenced in the manuscript.

2. Please update the completed 'Competing Interests' statement, including any COIs declared by your co-authors. If you have no competing interests to declare, please state "The authors have declared that no competing interests exist". Otherwise please declare all competing interests beginning with the statement "I have read the journal's policy and the authors of this manuscript have the following competing interests:"

3. Please remove the figures embedded in your manuscript file leaving only the separate figures files.

Additional Editor Comments (if provided):

The reviewers all believe this paper addresses an important topic. Two of the reviewers have expressed major concern with the methods, including description of what you have done, description of FHRM coverage, and the choice of analysis. Please give close attention to the methodological comments - particularly from Reviewer 5 who has published substantively on FHRM. Please address the comments as well as you can and if you feel any comments are unfair please leave comments to explain in the Comments to Editor section. I look forward to receiving your revised submission.

Reviewers' comments:

Reviewer's Responses to Questions

**Comments to the Author**

1. Does this manuscript meet PLOS Global Public Health’s publication criteria? Is the manuscript technically sound, and do the data support the conclusions? The manuscript must describe methodologically and ethically rigorous research with conclusions that are appropriately drawn based on the data presented.

Reviewer #1: Yes

Reviewer #2: Yes

Reviewer #3: No

Reviewer #4: Yes

Reviewer #5: Partly

2. Has the statistical analysis been performed appropriately and rigorously?

Reviewer #1: Yes

Reviewer #2: Yes

Reviewer #3: No

Reviewer #4: Yes

Reviewer #5: No

3. Have the authors made all data underlying the findings in their manuscript fully available (please refer to the Data Availability Statement at the start of the manuscript PDF file)?

Reviewer #1: Yes

Reviewer #2: Yes

Reviewer #3: Yes

Reviewer #4: Yes

Reviewer #5: Yes

4. Is the manuscript presented in an intelligible fashion and written in standard English?

Reviewer #1: Yes

Reviewer #2: Yes

Reviewer #3: No

Reviewer #4: Yes

Reviewer #5: Yes

5. Review Comments to the Author

Reviewer #1: First of all , I would like to thank the authors in making a strong effort doing a prospective cohort study around 12 hospitals in Nepal to make sure they find out the best care and the best approach for pregnant ladies in the birth suite.

Second, the study needs some minor revision from English and grammar perspective, for example" This study used data from a period of 18 months- instead you can write the data had been collected over a 18 months time frame between April 2017 and October 2018 "

Third, you included bar graph and P value chart, I would suggest including them in the body of the study to make the reader understands better the data instead of writing the at the end on the study.

Forth, the declaration is written at the end of the study, I would advise to write it before the introduction of the study

Reviewer #2: The title is good and it reflects the authors context of the problem phenomenon. The background information is substantial enough. The methodology is adequate, however, there was a comparison stated in the result aspect which the methodology didn't state if there is a comparison group to the cohort group.

The result is good. Data was rightly processed analysed. However, the logistic regression to establish the association was not clearly seen. There is a disconnect also between the title and the conclusion aspects. The major objective of the study was establish the association between fatal heart rate monitoring and intrapartum related death so focus should be on analysing data and use as evidence to accept or reject hypothesis that guides the study.

Reviewer #3: Summary:

The study aiming to assess the coverage of FHRM as per protocol and the association with intrapartum death in hospitals in Nepal is a highly concerning idea considering the addressed country, as well, recommendable to attempt due to its growing concern and lack of immediate preventive steps. Where, in one hand, the idea was contributive to the global research arena, the definitions and the statistical analyses of the findings may need to be sounder to be representative of a nation, on the other hand.

All in all, the initiative of such study is much appreciated and encouraged as the concept is really interesting, but a firm, clearer and understandable definition and presentation of the concept with proper supportive literature review along with clearer, brief and more appropriate method, analysis and apt data presentation may result in better outcome in future where the conclusion would be supportive to the elements elaborated in the manuscript.

Major essential revisions:

In method, the hospital selection criteria were not found, as well, the inclusion and exclusion criteria for selecting the participants were not backed up with supporting literatures by in texted citations which may create barriers in understanding the validation and lessens the chances of recreating it in a similar setting. How the randomization was assured at the time of sample selection was not mentioned. The data collection and management section was not clearly and elaborately explained. On what basis the data for FHRM variables and birth outcome were extracted from the medical records? As it’s a follow-up study, was the cohort being followed up on a daily basis or the data for the total period was collected at once? If the second one was applied, would it be considered as a prospective cohort or a retrospective cohort? Bit more detailed explanation of the data collection might come in handy to grasp the method. About the data management, was the data only cleaned using Survey Processing System (CS-Pro)? Moreover, was all the data managed at once using this software package and that also at the end of the follow up period or maintaining a basis (daily/weekly/monthly)? These information should be elaborative in a cohort study.

In measurements, the source of outcome measure definition should be mentioned. The absence of such, results in invalidation, hence stating the outcome variable of the current study undefined.

The main exposure variable definition was not supported by any citation where the WHO standard for the mentioned statement “Adherence to FHRM protocol during labor was deemed to be adequate when the FHR was monitored as per protocol at least every 30 minutes during the first stage of labor and at least every 15 minutes during the second stage of labor according to WHO standards” could be found.

Not a single mention of the non-exposed group, neither the definition of non-adherence to FHRM was found. Hence, the measurement for depicting sub-optimal FHRM is unstated and undefined.

None of the variables’ measures except for the “Obstetric complication upon admission” were validated with either reliable or any resources. Such actions could be addressed as invalidation of a manuscript.

The mention of significance level value which would be accepted could be mentioned. The ethical consideration section should be present in the method too.

In the flowchart, there was no mention of sample dropout, a mention of “no sample dropped out or died while following the cohort” might dispel any scope of measurement bias.

In result, mention of tests for descriptive analysis could be more appropriate and inferential statistical models e.g., bi-variate logistic regression may not be appropriate for descriptive statistics. By what type of tests “The bi-variate association between intrapartum related death with primary exposure-FHRM and other exposures” - was assessed? It should be stated clearly.

The objective did not mention of more than one outcome, therefore the mention of “multi-variate” association is not accepted as there’s a difference between multivariable and multivariate outcome.

Coming to the 3 models mentioned here, it says, “Model I was done by adjusting the primary exposure with antepartum related variable”. It would be helpful to assess the process if could know what the antepartum related variables exactly in this study are, as those were not found mentioned in the individual sub section under the method section, moreover, the antepartum deaths were one of the exclusion criteria.

“A total of 209 intrapartum stillbirths and 668 early neonatal deaths during the study period, with intrapartum related mortality rate of 10.5 per thousand deliveries. (Figure 1) The probability of intrapartum related mortality varied by hospital on a scale from 5 – 26 per 1000 birth. (Figure 2)” - Stating sentences like these within a para, is really unstructured and not actually depicting anything, it is near to suitable for a legend of a figure. Again, no mention of such values (209, 668) was observed in figure 1.

“More than 60% of the women delivering in Lumbini, Bheri, Koshi and Lumbini hospital had sub-optimal FHRM.” The data illustrated from figure 3 showed more than 75% of women delivering in Lumbini, Bheri having sub-optimal FHRM as opposed to Koshi having less than that indicating the need for better elaboration of data presentation.

Why assessing the odds by the variables’ crude values instead of observing by their adjusted values? This leaves the scope of confounding bias as the tentative confounders are not adjusted here in table 2, hence the results do not hold a great significance probably.

Besides, no notes under the tables mentioning abbreviation of terms, (e.g., aOR, cOR) and how the p-value was set as significant were found. Again, the contents of the table 2 was not labeled with their respective headings properly (e.g., what does “Yes, 37011 (44.2%), No, 46698 (55.8%) as the heading of table 1 or Alive (82,832, 99.0%), Intrapartum related death (877, 1.0%) as the heading of table 2” such headings mean?). Also, what are the percentage presented in the above-mentioned examples indicating? Are they prevalence? Or, are they presenting only the percent value of the distribution here?

A bit more emphasis could be given in the result and the method sections as these are two of the most important sections of a scientific report.

Minor essential revisions:

The introduction writing format lacked global perspective and strengthened justification. Introduction could be more sufficed. The study objective might be justified enough if it were cited with more evidences from related studies across the LMICs around of South Asian countries. Again, lack of global evidences from studies weakened both the purpose and background of the study. Also, elaborated information related to the main outcome could have been included.

In method, the necessity of five data sets and the function of MICE method should have been explained. The data created after the sensitivity analysis gathered in a supplementary table could have been included along with the submission, exemption in which does not really hold the statement any longer meaningful. STROBE, TRIPOD or CONSORT checklists should have been followed for the data reporting as its highly recommended.

In result, “Descriptive analysis of intrapartum related death by hospital and coverage of FHRM by hospitals”, a structured sentence opposed to this is much preferred.

Discretionary revisions:

Few grammatical corrections, uses of preposition and word selection may result in greater outcome for the readers’ understanding. In the end, unfortunately, to me, the conclusion is not adequately supported by the definition provided in the methods and strengthened by the data provided in some of the tables and figures. These sections could be revised.

Recommendations:

The study concept in itself is alarming, the large number of sample size is also a significant point, but the conceptual framework could be more suitable, brief and accurate by following the reporting guidelines of scientific report writing and the findings could be summarized along with the implementation ideas backing up the intervention applied.

Reviewer #4: Overall, I think the paper's aim is presented in a very clear manner, it also addresses an important issue, is written clearly, and presents important finding.

My comments are as follows:

-I would urge the authors to revise the definition of parity used in this paper (parity vs gravity) as at present their terminology is confusing. All definitions could be put in a table.

- It is plausible that the degree of prematurity (<28, 28-32,>32 weeks) could influence the outcome. Could the authors explain why it was used as binary rather than categorical.

- Could you please remove the p values from table 1, as hypothesis testing does not seem appropriate.

- Table 2, why was age group 26-30 used as baseline? The age group 19-25 makes more theoretical (less risk) and statistical (larger category/ more stable model) sense. Again, revise the use of parity terminology and consider preterm as categorial. Sex: M/F more appropriate.

-Could the authors conduct sub-group analysis for women who had complications during admission to help further stratify the risk (maternal vs fatal complications).

- Definition of who companions were would help contextualise the findings (family? doula?)

- Hart’s law of inverse care is very useful concept here, the discussion would benefit from few more sentences to explain this point better.

Reviewer #5: Title:

The title does not reflect the contents of the paper accurately

The title can be best written as “factors associated with poor adherence to intrapartum FHRM in relationship to intrapartum related deaths”

Abstract

To the best of my understanding: this is not a cohort study rather a “prospective observational study”. The time of follow up is only during labor and that cannot be termed at Cohort

There is term “admission” is consistently used but it is not specific where are the pregnant women admitted is it labor ward? Antenatal ward? Theatre or where? Authors need to be specific as admission to labor ward is different from admission to ANC ward for example

The term coverage is used vaguely throughout the document. It is better to be specific, use “proportional of FHRM monitored per protocol” or percentage of adherence. Coverage should not be used

The paper has two main aims and need to be clarified:

• Predictors of suboptimal FHRM

• Association of sub-optimal FHRM and intrapartum deaths

These two are not being specific in your paper

I am not a statistician, but I was trained to use the term “likelihood” with caution. In the abstract conclusion the term likelihood stands for “odds” and these two must not be used interchangeably. Likelihood is used when you are comparing the two models and you have a ratio with p-value comparing the models.

Introduction:

The first sentence is not clear please re-write

Overall, the introduction is too much focus on the literature from the same study group (papers referred) and Nepal. Authors did not take time to search for available evidence from other scientists across the world especially in Africa and specifically Tanzania, Uganda, and DR Congo where similar interventions are being implemented. I advise to re-write this section

Methods

Has NePeriQIP study been published (I think the answer is Yes). If ye,s don’t the authors see that these data have been published and if Not, then the evaluation of the project will be duplication of efforts?

Was this study described in the protocol or it just emerged?

What was the instrument used to measure FRH? This is important because different devices has different levels of accuracy

Why was GA be estimated by LNMP instead of adding more robust methods such as use of 1st trimester Ultrasound?

Pre-term labor; what if the GA was unknown, how was that established?

Statistical analysis:

Why was multi-level modelling done?

It is not clear in the manuscript how was this done and for what purpose

The use of this methods would benefit by adding a simple formula (stata syntax/code) within the text

Why was MAR used as the imputation methods and no other methods?

Results:

In the first paragraph, Why were the hospitals classified

Reference:

Most of the reference are from same study group. Authors are advised to seek evidence elsewhere to make their study robust

Figure 1:

Authors did imputation to counter missing values but there still missing data, why? Why was this missing information not imputed?

Figure 2 is missing

Figure 3: probability of intrapartum related deaths

Change label from probability estimate to “point estimates”

There is a lot of redundant information. Authors should decide either to use point estimates and CI in table or just the lines with point estimates and Cis embedded within but not both as same information is presented twice and this will confuse the reader

6. PLOS authors have the option to publish the peer review history of their article (what does this mean?). If published, this will include your full peer review and any attached files.

**Do you want your identity to be public for this peer review?** For information about this choice, including consent withdrawal, please see our Privacy Policy.

Reviewer #1: **Yes: **Kareem Haloub

Reviewer #2: **Yes: **Dr Adebayo

Reviewer #3: No

Reviewer #4: No

Reviewer #5: No

---

## [Decision Letter · Decision Letter 1]

2 Mar 2022

Factors associated with poor adherence to intrapartum fetal heart monitoring in relationship to intrapartum related death: a prospective cohort study

PGPH-D-21-00535R1

Dear Dr. KC,

We are pleased to inform you that your manuscript 'Factors associated with poor adherence to intrapartum fetal heart monitoring in relationship to intrapartum related death: a prospective cohort study' has been provisionally accepted for publication in PLOS Global Public Health.

Best regards,

Hamish R Graham

Academic Editor

Thank you for your resubmission and for addressing the reviewer comments in detail.

Reviewer Comments (for reference):

Reviewer's Responses to Questions

**Comments to the Author**

1. If the authors have adequately addressed your comments raised in a previous round of review and you feel that this manuscript is now acceptable for publication, you may indicate that here to bypass the “Comments to the Author” section, enter your conflict of interest statement in the “Confidential to Editor” section, and submit your "Accept" recommendation.

Reviewer #5: All comments have been addressed

2. Does this manuscript meet PLOS Global Public Health’s publication criteria? Is the manuscript technically sound, and do the data support the conclusions? The manuscript must describe methodologically and ethically rigorous research with conclusions that are appropriately drawn based on the data presented.

Reviewer #5: Yes

3. Has the statistical analysis been performed appropriately and rigorously?

Reviewer #5: Yes

4. Have the authors made all data underlying the findings in their manuscript fully available (please refer to the Data Availability Statement at the start of the manuscript PDF file)?

Reviewer #5: Yes

5. Is the manuscript presented in an intelligible fashion and written in standard English?

Reviewer #5: Yes

6. Review Comments to the Author

Reviewer #5: Recommended for publication

7. PLOS authors have the option to publish the peer review history of their article (what does this mean?). If published, this will include your full peer review and any attached files.

**Do you want your identity to be public for this peer review?** For information about this choice, including consent withdrawal, please see our Privacy Policy.

Reviewer #5: No
